# A retrospective review on antibiotic use in acute watery diarrhea in children in a tertiary care hospital of Karachi, Pakistan

Sonia Qureshi[1]⊕*, Shahzadi Resham[1]⊕, Mariam Hashmi[2], Abdullah B. Naveed[3], Zoya Haq[4], Syed Asad Ali[1]

1 Department of Pediatrics and Child Health, Aga Khan University, Karachi, Pakistan, 2 Graduate Medical Student, Aga Khan University, Karachi, Pakistan, 3 Medical Student, Aga Khan University, Karachi, Pakistan, 4 Medical Student, Liaquat National Hospital and Medical College, Karachi, Pakistan

⊕ These authors contributed equally to this work.
* sonia.qureshi@aku.edu

## Abstract

### Introduction

Responsible for at least one in nine pediatric deaths, diarrheal diseases are the leading, global cause of death. Further abetted by improper antibiotic use in a hospital setting, children with acute watery diarrhea can see prolonged hospital stays, and unwanted adverse effects such as antibiotic resistance. Hence, this study is aimed to identify the association between antibiotic usage for the treatment of acute watery diarrhea in children, and the impact this line of management has on the duration of their hospital stay.

### Methods

A retrospective review was conducted at the department of Pediatric of Aga Khan University Hospital (AKUH) in Karachi. A total of 305 records of children aged 6 months to 5 years who were admitted with a diagnosis of acute watery diarrhea from June 2017 –December 2018 was screened, of which 175 fulfilled the eligibility criteria. A predesigned questionnaire was used to collect demographic information, comorbidities, and clinical features, severity of dehydration, clinical examination, treatment received, and laboratory investigations. The primary outcome of this study was the length of hospital stay measured against the number of hours a child stayed in hospital for treatment of acute watery diarrhea. The statistical analysis was carried out using STATA version 14 to reach conclusive results.

### Results

175 patients presented with acute watery diarrhea, out of which 106 (60.6%) did not receive antibiotics. The median (IQR) age of the group that did not receive antibiotics was 12.0 (12.0) months compared to 15.0 (12.0) months for the group that did receive antibiotics. In both groups, there were more males than females, less than 15% of the patients were severely malnourished (WHZ score -3SD) and less than 10% of the patients were severely dehydrated. The median (IQR) length of hospital stay (hours) was 32.0 (19.0) respectively

**Data Availability Statement:** All relevant data are within the manuscript.

**Funding:** The author(s) received no specific funding for this work.

**Competing interests:** The authors have declared that no competing interests exist.

for the group that did not receive antibiotic and 41.0 (32.0) for the group that did receive antibiotic therapy. The expected length of hospital stay for the group that received antibiotic therapy was 0.22 hours higher than the group that did not. Finally, as compared to females, hospital stay for males was longer by 0.25 hours.

## Conclusion

In conclusion, antibiotic use was associated with a prolonged hospital stay in children with acute watery diarrhea as compared to children who did not receive antibiotics. Large scale robust prospective studies are needed to establish this association using this observational data.

## Introduction

Diarrheal diseases remain a foremost public health burden in children under 5 years of age especially in low-resource settings. Globally, there are nearly 1.7 billion cases of childhood diarrheal disease each year of which 36 million can be characterized as moderate or severe. 26% (9.3 million) cases were estimated to arise in Southeast Asia [1]. Every year diarrhea kills around 525,000 children under 5 years of age [2] and 25% of deaths in young children living in Africa and South-east Asia are due to acute gastroenteritis [3–5].

Acute watery diarrhea is a clinical syndrome defined by an increase in stool frequency ($\geq 3$ loose or watery stools in 24 hours) or loose bowel movements that exceeds the child's normal number of daily bowel movements, by two or more, with or without vomiting, fever, and abdominal pain [6–9]. Oral Rehydration Therapy (ORS) is the first line treatment for acute gastroenteritis as suggested by the American Academy of Pediatrics (AAP), Centers for Disease Control and Prevention (CDC), European Society for Pediatric Gastroenterology and Nutrition, and the World Health Organization (WHO) [8,10–12]. In developing countries, guidelines suggest that non-bloody diarrhea must be managed with fluids only, whereas dysentery can be managed with antibiotics. Antibiotics are primarily given to reduce the intensity and duration of diarrheal symptoms. Since the yield of stool culture is low, the decision to start antimicrobial therapy for acute diarrhea must be made on clinical grounds, and the choice of the antimicrobial agent has to be made empirically. As outlined by the Global Enteric Multi-center Study (GEMS), *rotavirus* is the leading cause of watery diarrhea in children aged 0–23 months, followed by *Shigella* and *Cryptosporidium*. However, in older children aged 23–59 months, the commonest cause of watery diarrhea is *Shigella*. Moderate-to-severe cases of diarrhea are caused by rotavirus, *Cryptosporidium*, *Enterotoxigenic Escherichia coli* and *Shigella* [4]. When Cholera, Shigellosis, dysentery due to Campylobacter and non-typhoidal Salmonellosis cause persistent diarrhea, and sharply decrease host immunity due to severe malnutrition, chronic disease, or lymphoproliferative disorders, an antibiotic regimen is highly indicated [13]. Bacterial etiology is one of the commonest causes of acute gastroenteritis, however, unless severe, prescription of antibiotic therapy is not recommended in children. The judicious use of antibiotics is especially warranted due to high rates of antibiotic resistance in developing countries [14–16]. Additionally, antibiotics have several side effects, especially fluoroquinolones, which increase the risk of peripheral neuropathy in the pediatric population [17]. Therefore, it is important to study the effect of using antibiotics in hospitalized children with acute gastroenteritis so treatment outcomes like duration of symptoms and length of hospital stay can be measured. This enables physicians to use antibiotic therapy more wisely and cautiously. This

study aims to determine the association between antibiotic use and length of hospital stay in children presenting with acute watery diarrhea in a tertiary care hospital. It also examines the relationship between antibiotic use, number of diarrheal episodes within 36 hours of presentation and the requirement for intravenous rehydration after 24 hours of presentation in children admitted with acute watery diarrhea. The data accumulated is much needed to develop local standard guidelines and answer policy-relevant questions for the use of antibiotics in children with acute watery diarrhea.

## Methods

A retrospective review was conducted at the department of Pediatric of Aga Khan University Hospital (AKUH) in Karachi. AKUH is a 740-bed private tertiary care academic hospital located in the metropolitan city of Karachi. Medical records of children aged 6 months to 5 years who were admitted with a diagnosis of acute watery diarrhea i.e. children presented with diarrhea of ≤ 14 days duration with ≥3 passages of loose watery stools within 24 hours containing no blood from June 2017 –December 2018 were reviewed retrospectively. Patients with dysentery, a history of antibiotic use in the last 5 days of hospitalization, chronic diarrhea or other chronic illnesses, and intussusception were excluded from the study. Immunocompromised children or those who left against medical advice (LAMA) within a few hours of hospital admission were also excluded. A total of 305 records with the diagnosis of acute watery diarrhea were screened and 175 records that fulfilled the eligibility criteria were included in the study (Fig 1).

Data on demographic information, comorbidities, clinical features like duration of diarrhea, duration of fever (if present), severity of dehydration, clinical examination, treatment received, and laboratory investigations was collected on a predesigned questionnaire. Length of hospital stay was the primary outcome measured as number of hours a child stayed in hospital for the treatment of acute watery diarrhea. Other outcome variables were the need for intravenous rehydration after 24 hours of hospital admission and number of diarrheal episodes within 36 hours of admission. Use of antibiotic was the exploratory variable and was defined as a child admitted with acute watery diarrhea who received antibiotic within 24 hours of presentation and was continued during the hospital stay.

Statistical analysis was carried out using STATA version 14. Frequencies with percentages were reported for categorical variables such as gender, weight for height/length Z- scores, status of dehydration, use of probiotics or zinc sulfate etc. For continuous variables such as age, duration of diarrhea before admission, number of diarrheal episodes, duration of fever and length of hospital stay (hours), mean ± SD or median (IQR) were reported.

A linear regression model was used to assess the linear relationship of independent variables individually with the length of hospital stay (hours). In univariate analysis, p-value ≤0.25 was considered significant. Results were reported as coefficients with 95% CI. After considering multicollinearity issue was fit a multivariable linear regression model using a stepwise approach, keeping the level of significance 5%. Plausible biological interactions, confounders, outliers, and influential data were also checked before evaluating goodness of fit of the model. Poisson regression was used to analyze the relationship of number of diarrheal episodes with antibiotic use and other independent variables. The results were reported as coefficients with 95% confidence interval. To assess need for intravenous rehydration after 24 hours of presentation, cox proportional hazard algorithm was used to determine the association with independent variables. Results were reported as PR with 95% CI. p- values of ≤0.25 and ≤ 0.05 were considered significant for univariate and multivariable analysis, respectively. This study was

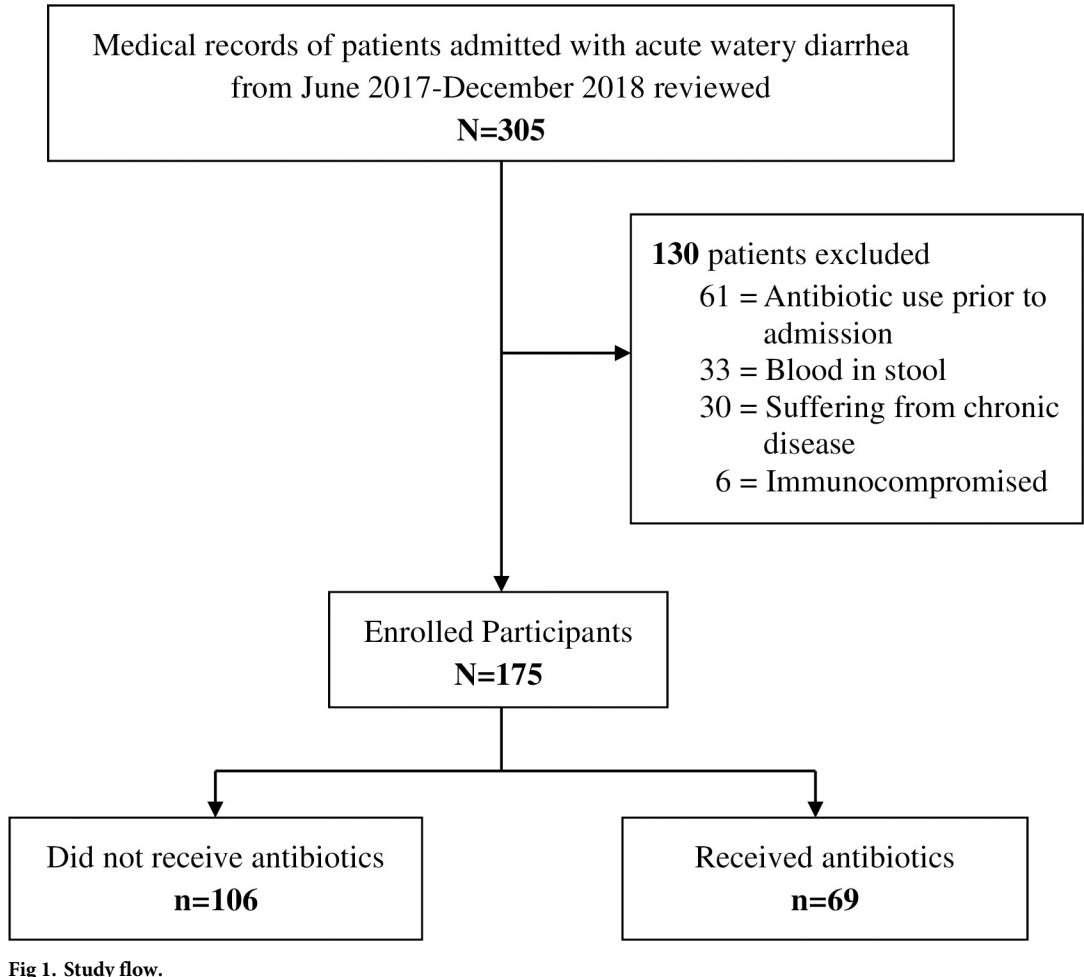

**Fig 1. Study flow.**

exempted from the Ethical Review Committee (ERC) of Aga Khan University Hospital (**ERC # 2019-0972-2452**).

## Results

175 patients with acute watery diarrhea were included in the study of which 106 (60.6%) did not receive antibiotics. Among those who received antibiotics, 44 (63.8%) children received cephalosporin (ceftriaxone or cefixime), 20 (28.9%) received ciprofloxacin, 2 (2.9%) received azithromycin and 1 (1.4%) each received ampicillin, ceftriaxone followed by either ciprofloxacin or azithromycin. The median (IQR) duration of antibiotics given was 2.0 (1.0) days. The median (IQR) age of the group that did not receive antibiotics was 12.0 (12.0) months compared to 15.0 (12.0) months for the group that did receive antibiotics. In both groups, there were more males than females, less than 15% of the patients were severely malnourished (WHZ score -3SD) and less than 10% of the patients were severely dehydrated. More than 65% of the patients in both groups had a history of fever with the median (IQR) duration of fever in the antibiotic group being 2.0 (2.0) days compared to 2.0 (2.0) in the other group. Among non-antibiotic group, 73 stool cultures were performed, of which one showed growth of *Salmonella Typhimurium*, however, 5 out of 35 stool cultures from the antibiotic users showed

**Table 1. Descriptive statistics of children with acute watery diarrhea in a tertiary care hospital, Karachi.**

| S.No | Variables | | No Antibiotics (N = 106) n (%) | Antibiotics (N = 69) n (%) |
|------|-----------|---|--------------------------------|----------------------------|
| 1 | Age (months) Median (IQR) | | 12.0 (12.0) | 15.0 (12.0) |
| 2 | Gender | Male | 77 (72.6) | 39 (56.5) |
| 3 | Weight for Height/Length (Z-scores) | -1 SD | 80 (75.5) | 49 (71.0) |
| | | -2 SD | 15 (14.2) | 11 (15.9) |
| | | -3 SD | 11 (10.4) | 9 (13.0) |
| 4 | Severity of dehydration | No | 35 (33.0) | 31 (44.9) |
| | | Some | 69 (65.1) | 33 (47.8) |
| | | Severe | 2 (1.9) | 5 (7.3) |
| 5 | Duration of diarrhea before admission (days) Median (IQR) | | 3.0 (2.0) | 2.0 (2.0) |
| 6 | No. of diarrheal episodes Median (IQR) | | 6.0 (8.0) | 5.0 (5.0) |
| 7 | History of fever | | 73 (68.9) | 51 (73.9) |
| 8 | Duration of fever (days) Median (IQR) | | 2.0 (2.0) | 2.0 (2.0) |
| 9 | Need of intravenous rehydration after 24 Hours of presentation | | 30 (28.3) | 28 (40.6) |
| 10 | Length of hospital stay (hours) Median (IQR) | | 32.0 (19.0) | 41.0 (32.0) |
| 11 | Use of probiotic | | 58 (54.7) | 32 (46.4) |
| 12 | Use of zinc sulphate | | 68 (64.2) | 37 (53.6) |
| 13 | Presence of ≥ 20 pus cells in stool DR | | 8/48 (16.7) | 11/43 (25.6) |
| 14 | Stool culture positive | | 1/73 (1.4) | 6/35 (17.1) |

growth of *Vibrio Cholerae*, *Salmonella species*, *Salmonella group B*, *Campylobacter* and 1 stool culture showed growth of 2 bacterial species (*Vibrio Cholerae* and *Campylobacter*) (Table 1).

The median (IQR) length of hospital stay (hours) was 32.0 (19.0) and 41.0 (32.0) respectively for the group that did not receive antibiotic and the group that received antibiotic. After adjusting for other co-variates, the expected length of hospital stay for the antibiotic group was 0.22 hours higher than the group that did not receive antibiotics (95% CI = 0.02, 0.43, p-value = 0.03). Males were also expected to have a longer hospital stay by 0.25 hours (95% CI: 0.02, 0.48 hours, p-value = 0.04) compared to females (Table 2).

The median (IQR) number of diarrheal episodes for the group that did not receive antibiotics was 6.0 (8.0) compared to 5.0 (5.0) episodes for the group that did receive antibiotics. We observed that with every 1 month increase in age, the number of diarrheal episodes decreased by 0.06 episodes (95% CI: -0.11, -0.01, p-value = 0.02 0.07) after controlling for other factors (Table 3). 30/106 (28.3%) of the children who did not receive antibiotics required IV rehydration after 24 hours compared to the 28/69 (40.6%) children from the group that received antibiotics. Children who received antibiotics were 1.62 times (95% CI: 0.96, 2.73, p-value = 0.07) as likely required the IV rehydration after 24 hours compared to children who did not receive antibiotics on adjusted analysis. The strength of this association was however not statistically significant (Table 4).

## Discussion

In this study, the effect of antibiotic use during acute watery diarrhea on treatment outcomes for patients in a developing country was assessed. By identifying factors that affect treatment outcomes clinicians may be enabled to treat paediatric gastroenteritis more effectively, especially in resource-limited settings.

**Table 2. Unadjusted and adjusted analysis of clinical characteristics and length of hospital stay (hours) in children presented with acute watery diarrhea in a tertiary care hospital.**

| S. No | Variables | | Unadjusted | | Adjusted | |
|---|---|---|---|---|---|---|
| | | | Coefficient (95% CI) | p- value | Coefficient (95% CI) | p-value |
| 1 | Use of Antibiotic | Yes | 0.21 (0.04, 0.38) | 0.02 | 0.22 (0.02, 0.43) | 0.03 |
| | | *No | 1 | | 1 | |
| 2 | Age (months) | | -0.008 (-0.02, 0.38) | 0.06 | -0.01 (-0.02, 0.00) | 0.04 |
| 3 | Gender | Male | 0.18 (-0.01,0.36) | 0.06 | 0.25 (0.02, 0.48) | 0.04 |
| | | *Female | 1 | - | 1 | |
| 4 | Weight for Height/Length (Z-scores) | *Normal | 1 | - | - | - |
| | | Moderate Acute Malnutrition (-2.99 to -2) | -0.13 (-0.36,0.11) | 0.23 | - | - |
| | | Severe Acute Malnutrition (-5 to -3) | 0.16 (-0.16,0.48) | 0.32 | - | - |
| 5 | Severity of Dehydration | *No | 1 | | 1 | |
| | | Some | -0.13 (-0.30,0.05) | 0.15 | -0.18 (-0.39,0.03) | 0.09 |
| | | Severe | -0.17 (-0.62, 0.27) | 0.45 | -0.27 (-0.64, 0.09) | 0.15 |
| 6 | Duration of diarrhea before admission (days) | | 0.015 (-0.03,0.06) | 0.48 | - | - |
| 7 | History of fever | Yes | -0.021 (-0.19,0.16) | 0.82 | - | - |
| | | *No | 1 | - | - | - |
| 8 | Duration of fever (days) | | 0.03 (-0.01, 0.08) | 0.17 | 0.01 (-0.03, 0.06) | 0.6 |
| 9 | No. of diarrheal episodes | | 0.06 (0.04, 0.07) | 0.001 | 0.05 (0.03, 0.06) | 0.001 |
| 10 | Intravenous rehydration after 24 hours of presentation | Yes | 0.53 (0.38, 0.68) | 0.001 | 0.45 (0.26, 0.63) | 0.001 |
| | | *No | 1 | - | - | - |
| 11 | Use of probiotic | Yes | -0.03 (-0.19, 0.14) | 0.71 | - | - |
| | | *No | 1 | | - | - |
| 12 | Use of zinc sulphate | Yes | -0.02 (-0.18, 0.15) | 0.85 | - | - |
| | | *No | 1 | | - | - |

* Reference category.

In this cohort, almost 39% (n = 69) of children were treated with antibiotics. Among those who received antibiotics, majority (n = 44, 63.8%) of them received cephalosporins followed by (n = 20, 28.9%) fluoroquinolones. A study by Kotwani et al stated that only 23% of children with diarrhea in government hospitals were prescribed antibiotics in contrast to 51.5% of children in private hospitals [18]. In this study, we found the same pattern however; there was no concrete evidence as to why this result was achieved. Inappropriate use of antibiotics is a major occurrence in most countries. This is affirmed by an Indian study that presented deviation from the WHO protocol in 78.4% of cases with improper use of antibiotics taking place in 12.2% of cases [19].

Patients who were given antibiotics had a longer length of stay (measured in hours) {Median (IQR), 41.0 (32.0)} as compared to patients not given antibiotics {Median (IQR), 32.0 (32.0)}. This might be because more children (7.3%) in the antibiotic group presented with severe dehydration compared to the non-antibiotic group (1.9%). These results are comparable to a study which found that antibiotic use increased the length of stay of children at the hospital [20]. In resource-limited countries, lack of infrastructure can play a role in prolonged hospital stays since patients find it more tedious to move back and forth for treatment. Additionally, antibiotics are only used empirically when patients present with severe diarrheal disease or have clinical signs and symptoms that warrant antibiotic use. It makes sense that these patients on average would stay in the hospital longer than those who did not meet the

**Table 3. Unadjusted and adjusted analysis of clinical characteristics and number of diarrheal episodes within 36 hours of presentation in children presented with acute watery diarrhea in a tertiary care hospital.**

| S. No | Variables | | Unadjusted | | Adjusted | |
|---|---|---|---|---|---|---|
| | | | Coefficient (95% CI) | p- value | Coefficient (95% CI) | p-value |
| 1 | Use of Antibiotic | Yes | -0.07 (-0.12, -0.03) | 0.46 | - | - |
| | | *No | 1 | - | - | - |
| 2 | Age (months) | | -0.07 (-0.12, -0.03) | 0.002 | -0.06 (-0.11, -0.01) | 0.02 |
| 3 | Gender | Male | 0.68 (-0.76, 2.13) | 0.35 | - | - |
| | | *Female | 1 | - | - | - |
| 4 | Weight for Height/Length (Z-scores) | *Normal | 1 | - | - | - |
| | | Moderate Acute Malnutrition (-2.99 to -2) | 0.05 (-1.92, 2.03) | 0.96 | - | - |
| | | Severe Acute Malnutrition (-5 to -3) | 0.238 (-1.97,2.45) | 0.83 | - | - |
| 5 | Severity of Dehydration | *No | 1 | - | 1 | - |
| | | Some | 1.55 (0.22, 2.87) | 0.02 | 1.24 (-0.13, 2.60) | 0.08 |
| | | Severe | 1.88 (-1.91, 5.67) | 0.33 | -0.27 (-0.64, 0.09) | 0.39 |
| 6 | Duration of diarrhea before admission (days) | | 0.05 (-0.26, 0.37) | 0.75 | - | - |
| 7 | History of fever | Yes | 0.53 (-0.89, 1.94) | 0.47 | - | - |
| | | | -0.021 (-0.19,0.16) | | | |
| | | *No | 1 | - | - | - |
| 8 | Duration of fever (days) | | 0.02 (-0.35, 0.38) | 0.93 | - | - |
| 9 | Intravenous rehydration after 24 hours of presentation | Yes | 2.82 (1.38, 4.27) | 0.001 | 0.08 (0.03, 0.13) | 0.003 |
| | | *No | 1 | | | |
| 10 | Length of hospital stay (hours) | | 0.09 (0.04, 0.13) | 0.001 | 0.71 (-1.00, 2.42) | 0.42 |
| 11 | Use of probiotic | Yes | 0.35 (-0.98,1.68) | 0.61 | - | - |
| | | *No | 1 | - | - | - |
| 12 | Use of zinc sulphate | Yes | -0.01 (-1.36, 1.35) | 0.99 | - | - |
| | | *No | 1 | - | - | - |

*Reference category.

criteria for antibiotic use. A study carried out in Ghana also reported that an increased severity of dehydration at presentation lead to a prolonged hospital stay [21]. However, this study did not mimic similar results and this outcome is most probably reliant on the small sample size. A similar finding was portrayed by a study conducted in the United States to see the variability of antibiotic use in pediatric care concluded that hospitals that exposed a higher proportion of patients to antibiotics also used more days of therapy [22]. The authors found this conclusion surprising since it was contrary to the popular belief that hospitals where patients are given empirical antibiotic therapy see drug use and hospital stay for a shorter time. It is reasonable to assume that with increased severity of dehydration, hospital stays would be longer to correct the hydration status. Established factors that affect the length of stay include fever at the time of presentation and compromised base line nutrition status. From the finding pertaining to males having longer stay time in the hospital as compared to females, it can be concluded that 74/116 (63.7%) of male patients had signs of dehydration on presentation compared to females 35/59, (59.3%) or may be this observation is due to the fact that in a developing country like Pakistan, where illiteracy is one of the biggest social dilemmas, families are more prone to insist on prolonged admission of their male heirs as compared to females. This can be a major contributing factor in the difference in results, especially when viewed as a component of gender inequality [23,24].

**Table 4. Unadjusted and adjusted analysis of clinical characteristics and need of intravenous rehydration after 24 hours of presentation in children presented with acute watery diarrhea in a tertiary care hospital.**

| S. No | Variables | | Unadjusted | | Adjusted | |
|---|---|---|---|---|---|---|
| | | | Crude PR (95% CI) | p- value | Adj PR (95% CI) | p-value |
| 1 | Use of Antibiotic | Yes | 1.43 (0.86, 2.40) | 0.17 | 1.62 (0.96, 2.74) | 0.07 |
| | | *No | 1 | - | - | - |
| 2 | Age (months) | | 0.96 (0.92, 0.99) | 0.01 | 0.96 (0.92, 0.99) | 0.01 |
| 3 | Gender | Male | 1.34 (0.75, 2.38) | 0.33 | - | - |
| | | *Female | 1 | - | - | - |
| 4 | Weight for Height/Length (Z-scores) | *Normal | 1 | - | - | - |
| | | Moderate Acute Malnutrition (-2.99 to -2) | 0.92 (0.43, 1.96) | 0.84 | - | - |
| | | Severe Acute Malnutrition (-5 to -3) | 1.05 (0.47, 2.33) | 0.91 | - | - |
| 5 | Severity of Dehydration | *No | 1 | - | - | - |
| | | Some | 1.73 (0.95, 3.12) | 0.07 | 1.24 (-0.13, 2.60) | 0.08 |
| | | Severe | 1.89 (0.55, 6.51) | 0.32 | 1.68 (0.48, 5.84) | 0.41 |
| 6 | Duration of diarrhea before admission (days) | | 0.96 (0.84, 1.09) | 0.49 | - | - |
| 7 | History of fever | Yes | 1.18 (0.66, 2.12) | 0.58 | - | - |
| | | *No | 1 | - | - | - |
| 8 | Duration of fever (days) | | 0.96 (0.83, 1.12) | 0.62 | - | - |
| 9 | No. of diarrheal episodes | | 1.09 (1.03, 1.14) | 0.002 | 1.04 (0.98, 1.10) | 0.16 |
| 10 | Length of hospital stay (hours) | | 1.01 (1.01, 1.02) | 0.001 | 1.01 (1.00, 1.02) | 0.003 |
| 11 | Use of probiotic | Yes | 1.16 (0.69, 1.95) | 0.57 | - | - |
| | | *No | 1 | - | - | - |
| 12 | Use of zinc sulphate | Yes | 0.77 (0.46, 1.28) | 0.31 | - | - |

*Reference category.

Recent guidelines do not endorse the use of antibiotics for acute watery diarrhea in pediatric population as most cases have a viral etiology and are therefore self-limiting. Exceptions include diarrhea caused by cholera, shigellosis, campylobacteriosis, non-typhoidal salmonellosis, and diarrhea in immunosuppressed patients [25]. Similarly, our data corroborated the growth of *Vibrio Cholerae*, *Salmonella species*, *Salmonella group B*, *Campylobacter* in 5 out of 35 stool cultures and 1 stool culture showed growth of 2 bacterial species (*Vibrio Cholerae* and *Campylobacter)*. In developing countries however, guidelines state that the presence of blood in stool must be checked before prescribing antibiotics. Non-bloody diarrhea should only be managed using fluids while management of dysentery requires antibiotic incorporation in the treatment regimen [26,27].

Whilst the results of our study are consistent with past literature both nationally and internationally, its biggest shortcoming lay in the fact that an etiology could not be established for each case. Another limitation was the retrospective nature of our study. In contrast, the lack of literature about the expected length of stay for acute gastroenteritis admissions and its management in hospital settings of Pakistan and other developing countries is scarce. The data generated from this study may set the ground as a primary study generating hypotheses to be further studied by larger, more expensive prospective studies to evaluate the need of antibiotics and its involvement in regional guidelines.

## Conclusion

In conclusion, antibiotic use was associated with a prolonged hospital stay in children with acute watery diarrhea compared to children who did not get antibiotics. Further large scale

robust prospective studies are required to establish this association and to determine whether or not regional guidelines need to be altered to account for different scenarios in which antibiotic use might be indicated.

## Author Contributions

**Conceptualization:** Sonia Qureshi, Shahzadi Resham.

**Data curation:** Sonia Qureshi.

**Investigation:** Mariam Hashmi, Abdullah B. Naveed.

**Methodology:** Mariam Hashmi, Abdullah B. Naveed.

**Supervision:** Sonia Qureshi.

**Writing – original draft:** Sonia Qureshi, Shahzadi Resham, Mariam Hashmi, Abdullah B. Naveed, Zoya Haq.

**Writing – review & editing:** Sonia Qureshi, Mariam Hashmi, Syed Asad Ali.

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
