## [Decision Letter · Decision Letter 0]

9 Mar 2021

PONE-D-21-04700

Association between antibiotic use and length of hospital stay among children presented with acute watery diarrhea in a tertiary care hospital, Karachi- A Cross-sectional Study

PLOS ONE

Dear Dr. Qureshi,

Thank you for submitting your manuscript to PLOS ONE. After careful consideration, we feel that it has merit but does not fully meet PLOS ONE’s publication criteria as it currently stands. Therefore, we invite you to submit a revised version of the manuscript that addresses the points raised during the review process.

Please revise accordingly.

We look forward to receiving your revised manuscript.

Kind regards,

Academic Editor

PLOS ONE

Journal Requirements:

2. In your ethics statement in the Methods section and in the online submission form, please provide additional information about the data used in your retrospective study. Specifically, please ensure that you have discussed whether all data were fully anonymized before you accessed them.

3. Thank you for providing the date(s) when patient medical information was initially recorded. Please also include the date(s) on which your research team accessed the databases/records to obtain the retrospective data used in your study.

5. Please upload a copy of Figure 1, to which you refer in your text on page 6. If the figure is no longer to be included as part of the submission please remove all reference to it within the text.

Reviewers' comments:

Reviewer's Responses to Questions

**Comments to the Author**

1. Is the manuscript technically sound, and do the data support the conclusions?

Reviewer #1: Partly

Reviewer #2: Yes

2. Has the statistical analysis been performed appropriately and rigorously? 

Reviewer #1: Yes

Reviewer #2: Yes

3. Have the authors made all data underlying the findings in their manuscript fully available?

Reviewer #1: Yes

Reviewer #2: Yes

4. Is the manuscript presented in an intelligible fashion and written in standard English?

Reviewer #1: Yes

Reviewer #2: No

5. Review Comments to the Author

Reviewer #1: The present retrospective study aimed to study the association between antibiotic use and length of hospital stay among children with acute watery diarrhea in a Pakistani tertiary care hospital. This is an important topic with important clinical implications. However, the study has some serious limitations, particularly in the methodology.

• The authors should provide more details on the methods of medical records review. What is the system for medical recording in this hospital? Is it electronic or paper based? Is “acute watery diarrhea” an agreed diagnostic term or has a specific code to be recorded? Who reviewed the records (a single person? Two?)? Was there a double check or revision? ….etc

• Authors should provide the indications of hospital admission for cases included in this study, what were the guidelines for antibiotics prescription in acute watery diarrhea in this hospital (39.4% of children with acute watery diarrhea who have no dysentery, no chronic illness, and no immunocompromised status received antibiotics), what were the guidelines for giving intravenous fluids (as need for intravenous rehydration after 24 hours of hospital admission was one of the study outcomes, and what were the discharge criteria (as length of hospital stay is the primary outcome). Is it possible that children who received the antibiotics stayed for a longer time in the hospital in order to complete a certain course of antibiotics?

• Authors should provide methods for assessment of the severity of dehydration,

• The authors conclude that antibiotic use was associated with a prolonged hospital stay in children with acute watery diarrhea. They report that the expected length of hospital stay for the antibiotic group was 0.22 hours higher than the group that did not receive antibiotics. However, even if this finding is statistically significant, I am concerned about the clinical importance with this small effect size (0.22 hours).

• In the discussion, authors should explain the possible reasons/mechanisms for their findings, such as the association of antibiotic use and increased length of hospital stay and the association between male gender and longer hospital of stay.

• The authors incorrectly use “cross-sectional study” instead of “retrospective study” in the title and methods.

Reviewer #2: Thank you for allowing me to review this important manuscript which is concerned with one of the most important diseases in children less than 5 years of age which represents a considerable cause of childhood mortality

The Language needs to be more polished and correction of some grammatical errors is essential

6. PLOS authors have the option to publish the peer review history of their article (what does this mean?). If published, this will include your full peer review and any attached files.

Reviewer #1: **Yes: **Elsayed Abdelkreem

Reviewer #2: No

---

## [Author Response · Author response to Decision Letter 0]

6 May 2021

Comments Responses

Reviewer#1: 

1. The authors should provide more details on the methods of medical records review. What is the system for medical recording in this hospital? Is it electronic or paper based? Is “acute watery diarrhea” an agreed diagnostic term or has a specific code to be recorded? Who reviewed the records (a single person? Two?)? Was there a double check or revision? …. etc 

Response:The system of recording at the hospital is paper-based and a digital record is also maintained for diagnosis from which data is extracted by using ICD codes.

We pulled out medical record numbers of all children admitted with acute gastroenteritis during the defined time period and those fulfilling the pre-defined clinical definition were enrolled.

2. Authors should provide the indications of hospital admission for cases included in this study, what were the guidelines for antibiotics prescription in acute watery diarrhea in this hospital (39.4% of children with acute watery diarrhea who have no dysentery, no chronic illness, and no immunocompromised status received antibiotics), what were the guidelines for giving intravenous fluids (as need for intravenous rehydration after 24 hours of hospital admission was one of the study outcomes, and what were the discharge criteria (as length of hospital stay is the primary outcome). Is it possible that children who received the antibiotics stayed for a longer time in the hospital in order to complete a certain course of antibiotics? 

Response: There is no local guideline. We usually follow the standard guidelines but decision to prescribe antibiotics depends upon the severity of underlying illness and at the physician’s discretion. Similarly IV rehydration is provided if child has severe loses or doesn’t tolerate orally.

3. Authors should provide methods for assessment of the severity of dehydration,

Response: The clinical assessment for severity of illness is based on skin turgor, eagerness to drink, sunken eyes and irritability as defined by WHO IMCI (Integrated Management of Childhood Illness) chart. We follow these four criteria for dehydration severity grading in our hospital. As it is a retrospective review, we collected documented information from medical record file of each patient at the time of presentation.

4. The authors conclude that antibiotic use was associated with a prolonged hospital stay in children with acute watery diarrhea. They report that the expected length of hospital stay for the antibiotic group was 0.22 hours higher than the group that did not receive antibiotics. However, even if this finding is statistically significant, I am concerned about the clinical importance with this small effect size (0.22 hours). 

Response:Thank you for highlighting this point. We understand this limitation. This may be because of small sample size.

5. In the discussion, authors should explain the possible reasons/mechanisms for their findings, such as the association of antibiotic use and increased length of hospital stay and the association between male gender and longer hospital of stay. This is really an important point which we missed in our manuscript to discuss. 

Response: Our findings indicated that antibiotic use was associated with a longer hospital stay. This may be because more children (7.3%) in the antibiotic group presented with severe dehydration compared to the non-antibiotic group (1.9%). Similarly 74/116 (63.7%) of male patients had signs of dehydration on presentation compared to females 35/59, (59.3%) or because in developing countries where illiteracy rate is high, families are more prone to insist on prolonged admission of their male heirs as compared to females. This can be a major contributing factor in the difference in results, especially when viewed as a component of gender inequality.

We have added the reasoning in the modified version of the manuscript (Line no: 215-216, 225-231, 234-241)

6. The authors incorrectly use “cross-sectional study” instead of “retrospective study” in the title and methods. 

Response: This query has been addressed in the updated manuscript and the title has been edited.

Reviewer#2: 

1. The Language needs to be more polished, and correction of some grammatical errors is essential 

Response: This query has been addressed by an in-depth analysis of the paper and corrections in the updated manuscript.

---

## [Decision Letter · Decision Letter 1]

9 May 2021

PONE-D-21-04700R1

Association between antibiotic use and length of hospital stay among children presented with acute watery diarrhea in a tertiary care hospital, Karachi- A Cross-sectional Study

PLOS ONE

Dear Dr. Qureshi,

Thank you for submitting your manuscript to PLOS ONE. After careful consideration, we feel that it has merit but does not fully meet PLOS ONE’s publication criteria as it currently stands. Therefore, we invite you to submit a revised version of the manuscript that addresses the points raised during the review process.

Please revise accordingly.

We look forward to receiving your revised manuscript.

Kind regards,

Academic Editor

PLOS ONE

Reviewers' comments:

Reviewer's Responses to Questions

**Comments to the Author**

1. If the authors have adequately addressed your comments raised in a previous round of review and you feel that this manuscript is now acceptable for publication, you may indicate that here to bypass the “Comments to the Author” section, enter your conflict of interest statement in the “Confidential to Editor” section, and submit your "Accept" recommendation.

Reviewer #3: (No Response)

Reviewer #4: (No Response)

2. Is the manuscript technically sound, and do the data support the conclusions?

Reviewer #3: No

Reviewer #4: No

3. Has the statistical analysis been performed appropriately and rigorously? 

Reviewer #3: No

Reviewer #4: Yes

4. Have the authors made all data underlying the findings in their manuscript fully available?

Reviewer #3: Yes

Reviewer #4: Yes

5. Is the manuscript presented in an intelligible fashion and written in standard English?

Reviewer #3: Yes

Reviewer #4: No

6. Review Comments to the Author

Reviewer #3: General comment:

The study aimed to investigate the association between length of hospitalization and antibiotic use in children with acute watery diarrhea. They found a 0.22-hour difference in length of stay. After revision, the overall quality increased. However, the major limitations were limited clinical significance and limited information because of retrospective setting.

Major comments:

1. P6, L112: Is it a cross-section study or retrospective study?

2. The indication for antibiotic use or admission should be provided to make the result or conclusion more convincing. Or this could be the major limitation in your study.

3. The association between antibiotics use and prolonged hospitalization was not explained well in the discussion. In different settings, antibiotic choices, different background, all variables will change. In limited information from limited cases, the association was not strong enough to make the conclusion. Try to find more variables, such as categories of antibiotics, pathogens, might be helpful.

4. I have the same question about the clinical significance of “0.22 hour” longer in hospitalization as previous reviewer stated. It has minimal drive for clinician to change their daily practice.

Minor comments:

1. Virus should not be italic or uppercase compared with bacteria. Ex: Diarrhea caused by “rotavirus” and “Campylobacter jejuni”

Reviewer #4: General comments:

The authors retrospectively reviewed the effects of antibiotics on the length of hospital stay in 175 children with acute watery diarrhea in a single hospital. They found that the expected length of hospital stays for children who received antibiotic therapy was 0.22 hours higher than that of children who did not. The conclude that antibiotic use was associated with a prolonged hospital stay in children with acute watery diarrhea compared to children who did not receive antibiotics.

General concerns:

1. Methods: The diagnosis criteria of acute watery diarrhea were not defined. The authors described only exclusion criteria. The authors did stool culture (Table 1. Stool culture positive rate). Please describe the inclusion criteria and the antibiotics category and duration.

2. Antibiotic therapy is not necessary for acute diarrhea in children because rehydration is the crucial treatment and symptoms resolve without specific therapy. Please describe the indication for antibiotics in this manuscript.

3. Methods, page 112: The phrase “A cross-sectional study was conducted at the department of Pediatric---” was not corrected in the revised manuscript.

4. Figure 1: Please describe the number of the enrolled participants in each group.

7. PLOS authors have the option to publish the peer review history of their article (what does this mean?). If published, this will include your full peer review and any attached files.

Reviewer #3: No

Reviewer #4: No

---

## [Author Response · Author response to Decision Letter 1]

12 May 2021

Comments Responses

Reviewer#3: 

1. P6, L112: Is it a cross-section study or retrospective study? 

Response: We are very sorry that we forgot to do this change in the study manuscript.We have done the corrections in the revised manuscript (P5, L103). Sorry for the inconvenience.

2. The indication for antibiotic use or admission should be provided to make the result or conclusion more convincing. Or this could be the major limitation in your study 

Response: There is no local hospital guideline. We usually follow the standard guideline. The common practice in our institute to prescribe antibiotics depends upon the severity of underlying illness and at the physician’s discretion. This is the basis of conducting this study to increase the awareness of our physicians that all should stick and follow the standard guidelines and if diverting from the pathway there must be some rationale of starting the antibiotics as in acute watery diarrhea there is no role of antibiotics. Similarly most common reasons for admission in our hospital are unable to tolerate feed, dehydration and parental anxiety. As this is the retrospective review this information is difficult to collect. This study helps us in developing a local hospital guideline and improving the knowledge of our HCWS so that everyone follow that standard pathway and in this way we can able to control the unnecessary use of antibiotics in our setting and in settings where such practice follows by disseminating the results of our study.

3. The association between antibiotics use and prolonged hospitalization was not explained well in the discussion. In different settings, antibiotic choices, different background, all variables will change. In limited information from limited cases, the association was not strong enough to make the conclusion. Try to find more variables, such as categories of antibiotics, pathogens, might be helpful.

Response: We have mentioned the categories of antibiotics prescribed in our manuscript. Very few stool cultures came out to be positive. The description of which is given in the write-up. We know that the yield of stool culture to detect a bug is very low particularly in the setting where self-medication and overuse of antibiotics are common practice in the community. We agree with you that this association is not strong enough to make conclusion. To answer this question we need more large scale studies which we recommended in our conclusion. This study will help in developing the background/rationale to design large scale multicenter studies/trials. We have also mentioned/modified our recommendation in the abstract and in main text (P3, L51-52, P19, L276-277)). We hope we have responded your query appropriately.

4. I have the same question about the clinical significance of “0.22 hour” longer in hospitalization as previous reviewer stated. It has minimal drive for clinician to change their daily practice Thank you for highlighting this point. 

Response: We have elaborated this point in comment 3. We understand this limitation. Therefore we recommend larger studies in our conclusion.

5. Virus should not be italic or uppercase compared with bacteria. Ex: Diarrhea caused by “rotavirus” and “Campylobacter jejuni” 

Response: Thank you for picking this up. We have made corrections as suggested (P4, L82). 

Reviewer#4: 

1. Methods: The diagnosis criteria of acute watery diarrhea were not defined. The authors described only exclusion criteria. The authors did stool culture (Table 1. Stool culture positive rate). Please describe the inclusion criteria and the antibiotics category and duration

Response: Inclusion criteria is elaborated in methods section (P5, L106-108). Antibiotics categories and median duration of antibiotics are mentioned in the text of result section (P6, L146-149). Details of positive stool cultures are mentioned in the text of result section (P8, L155-159).

2. Antibiotic therapy is not necessary for acute diarrhea in children because rehydration is the crucial treatment and symptoms resolve without specific therapy. Please describe the indication for antibiotics in this manuscript. 

Response: Same as response 2 mentioned in reviewer 3 comments.

3. Methods, page 112: The phrase “A cross-sectional study was conducted at the department of Pediatric---” was not corrected in the revised manuscript. 

Response: Thankyou so much for highlighting this. We have corrected it in the revised manuscript (P5, L103).

4. Figure 1: Please describe the number of the enrolled participants in each group. 

Response: We have added this information in the Figure. Kindly refer to the revised version.

---

## [Decision Letter · Decision Letter 2]

17 May 2021

PONE-D-21-04700R2

Association between antibiotic use and length of hospital stay among children presented with acute watery diarrhea in a tertiary care hospital, Karachi- Retrospective Study

PLOS ONE

Dear Dr. Qureshi,

Thank you for submitting your manuscript to PLOS ONE. After careful consideration, we feel that it has merit but does not fully meet PLOS ONE’s publication criteria as it currently stands. Therefore, we invite you to submit a revised version of the manuscript that addresses the points raised during the review process.

Please revise the Title to make it concise whereas informative.

We look forward to receiving your revised manuscript.

Kind regards,

Academic Editor

PLOS ONE

Journal Requirements:

Reviewers' comments:

Reviewer's Responses to Questions

**Comments to the Author**

1. If the authors have adequately addressed your comments raised in a previous round of review and you feel that this manuscript is now acceptable for publication, you may indicate that here to bypass the “Comments to the Author” section, enter your conflict of interest statement in the “Confidential to Editor” section, and submit your "Accept" recommendation.

Reviewer #3: All comments have been addressed

Reviewer #4: All comments have been addressed

2. Is the manuscript technically sound, and do the data support the conclusions?

Reviewer #3: Yes

Reviewer #4: Yes

3. Has the statistical analysis been performed appropriately and rigorously? 

Reviewer #3: Yes

Reviewer #4: Yes

4. Have the authors made all data underlying the findings in their manuscript fully available?

Reviewer #3: Yes

Reviewer #4: Yes

5. Is the manuscript presented in an intelligible fashion and written in standard English?

Reviewer #3: Yes

Reviewer #4: Yes

6. Review Comments to the Author

Reviewer #3: The authors tried hard to response all the issues I mentioned. Although some limitations and clinical questions exist, I expect to see more new results in the future. I have no more comment about the manuscript.

Reviewer #4: The authors have addressed the concerns.

7. PLOS authors have the option to publish the peer review history of their article (what does this mean?). If published, this will include your full peer review and any attached files.

Reviewer #3: No

Reviewer #4: **Yes: **Chung-Ming Chen

---

## [Author Response · Author response to Decision Letter 2]

19 May 2021

1. Please revise the Title to make it concise whereas informative 

Response: We have revised the title as suggested. Kindly refer to the revised manuscript.

---

## [Decision Letter · Decision Letter 3]

24 May 2021

PONE-D-21-04700R3

A retrospective review on antibiotic use in acute watery diarrhea in children in a tertiary care hospital, Karachi.

PLOS ONE

Dear Dr. Qureshi,

Thank you for submitting your manuscript to PLOS ONE. After careful consideration, we feel that it has merit but does not fully meet PLOS ONE’s publication criteria as it currently stands. Therefore, we invite you to submit a revised version of the manuscript that addresses the points raised during the review process.

Please revise accordingly and also the Title.  Readers have no idea of the location name.  Please revise the Title to make it concise and informative.

We look forward to receiving your revised manuscript.

Kind regards,

Academic Editor

PLOS ONE

Journal Requirements:

Reviewers' comments:

Reviewer's Responses to Questions

**Comments to the Author**

1. If the authors have adequately addressed your comments raised in a previous round of review and you feel that this manuscript is now acceptable for publication, you may indicate that here to bypass the “Comments to the Author” section, enter your conflict of interest statement in the “Confidential to Editor” section, and submit your "Accept" recommendation.

Reviewer #3: All comments have been addressed

Reviewer #4: All comments have been addressed

2. Is the manuscript technically sound, and do the data support the conclusions?

Reviewer #3: Yes

Reviewer #4: Yes

3. Has the statistical analysis been performed appropriately and rigorously? 

Reviewer #3: Yes

Reviewer #4: Yes

4. Have the authors made all data underlying the findings in their manuscript fully available?

Reviewer #3: Yes

Reviewer #4: Yes

5. Is the manuscript presented in an intelligible fashion and written in standard English?

Reviewer #3: Yes

Reviewer #4: Yes

6. Review Comments to the Author

Reviewer #3: The author have adequately addressed all the issues I mentioned. I have no further comment on this manuscript.

Reviewer #4: The authors have addressed the concerns. I have no further comments.

The authors have addressed the concerns. I have no further comments.

7. PLOS authors have the option to publish the peer review history of their article (what does this mean?). If published, this will include your full peer review and any attached files.

Reviewer #3: No

Reviewer #4: **Yes: **Chung-Ming Chen

---

## [Author Response · Author response to Decision Letter 3]

25 May 2021

1. Readers have no idea of the location name. Please revise the Title to make it concise and informative.

Response: We have added the location in title in the revised file. Kindly refer to the revised manuscript.

---

## [Decision Letter · Decision Letter 4]

1 Jun 2021

PONE-D-21-04700R4

A retrospective review on antibiotic use in acute watery diarrhea in children in a tertiary care hospital of Karachi, Pakistan.

PLOS ONE

Dear Dr. Qureshi,

Thank you for submitting your manuscript to PLOS ONE. After careful consideration, we feel that it has merit but does not fully meet PLOS ONE’s publication criteria as it currently stands. Therefore, we invite you to submit a revised version of the manuscript that addresses the points raised during the review process.

Please revise accordingly

We look forward to receiving your revised manuscript.

Kind regards,

Academic Editor

PLOS ONE

Journal Requirements:

Reviewers' comments:

Reviewer's Responses to Questions

**Comments to the Author**

1. If the authors have adequately addressed your comments raised in a previous round of review and you feel that this manuscript is now acceptable for publication, you may indicate that here to bypass the “Comments to the Author” section, enter your conflict of interest statement in the “Confidential to Editor” section, and submit your "Accept" recommendation.

Reviewer #3: All comments have been addressed

Reviewer #4: All comments have been addressed

Reviewer #5: (No Response)

2. Is the manuscript technically sound, and do the data support the conclusions?

Reviewer #3: Yes

Reviewer #4: Yes

Reviewer #5: Yes

3. Has the statistical analysis been performed appropriately and rigorously? 

Reviewer #3: Yes

Reviewer #4: Yes

Reviewer #5: Yes

4. Have the authors made all data underlying the findings in their manuscript fully available?

Reviewer #3: Yes

Reviewer #4: Yes

Reviewer #5: Yes

5. Is the manuscript presented in an intelligible fashion and written in standard English?

Reviewer #3: Yes

Reviewer #4: Yes

Reviewer #5: Yes

6. Review Comments to the Author

Reviewer #3: The author have addressed all the issues I mentioned. I have no further comment to this manuscript.

Reviewer #4: The authors have addressed all the raised concerns.

The authors have addressed all the raised concerns.

Reviewer #5: It is professionally written paper on one of important public health issue of developing countries. I suggest excluding table 3 and table 4 as both intravenous fluid and diarrheal episodes are not primary objectives of the report. Table 2 is sufficient to present association of clinical manifestations with hospital stay.

7. PLOS authors have the option to publish the peer review history of their article (what does this mean?). If published, this will include your full peer review and any attached files.

Reviewer #3: No

Reviewer #4: **Yes: **Chung-Ming Chen

Reviewer #5: **Yes: **Prof Sajid Soofi

---

## [Author Response · Author response to Decision Letter 4]

2 Jun 2021

Reviewer 5:

1. It is professionally written paper on one of important public health issue of developing countries. I suggest excluding table 3 and table 4 as both intravenous fluid and diarrheal episodes are not primary objectives of the report. Table 2 is sufficient to present association of clinical manifestations with hospital stay

Response: Thank you for your suggestion. We agree that table 3 and 4 are covering our secondary objectives. We request you to kindly keep it in the manuscript as they were our secondary objectives (Refer to highlighted part on page 5 of revised manuscript). These tables provide additional findings related to antibiotic use in acute watery diarrhea and number of diarrheal episodes and need of IV rehydration. They are adding further strength to our paper. If still reviewer wants to remove these tables we will remove them.

---

## [Decision Letter · Decision Letter 5]

11 Jun 2021

A retrospective review on antibiotic use in acute watery diarrhea in children in a tertiary care hospital of Karachi, Pakistan.

PONE-D-21-04700R5

Dear Dr. Qureshi,

We’re pleased to inform you that your manuscript has been judged scientifically suitable for publication and will be formally accepted for publication once it meets all outstanding technical requirements.

Kind regards,

Academic Editor

PLOS ONE

Additional Editor Comments (optional):

Reviewers' comments:

Reviewer's Responses to Questions

**Comments to the Author**

1. If the authors have adequately addressed your comments raised in a previous round of review and you feel that this manuscript is now acceptable for publication, you may indicate that here to bypass the “Comments to the Author” section, enter your conflict of interest statement in the “Confidential to Editor” section, and submit your "Accept" recommendation.

Reviewer #3: All comments have been addressed

Reviewer #4: All comments have been addressed

2. Is the manuscript technically sound, and do the data support the conclusions?

Reviewer #3: Yes

Reviewer #4: Yes

3. Has the statistical analysis been performed appropriately and rigorously? 

Reviewer #3: Yes

Reviewer #4: Yes

4. Have the authors made all data underlying the findings in their manuscript fully available?

Reviewer #3: Yes

Reviewer #4: Yes

5. Is the manuscript presented in an intelligible fashion and written in standard English?

Reviewer #3: Yes

Reviewer #4: Yes

6. Review Comments to the Author

Reviewer #3: The author have addressed all the issues I mentioned. I have no further comment to this manuscript.

Reviewer #4: The authors have fully addressed the reviewers concerns.

The authors have fully addressed the reviewers concerns.

7. PLOS authors have the option to publish the peer review history of their article (what does this mean?). If published, this will include your full peer review and any attached files.

Reviewer #3: No

Reviewer #4: **Yes: **Chung-Ming Chen

---

## [Editor Report · Acceptance letter]

6 Jul 2021

PONE-D-21-04700R5 

A retrospective review on antibiotic use in acute watery diarrhea in children in a tertiary care hospital of Karachi, Pakistan. 

Dear Dr. Qureshi:

I'm pleased to inform you that your manuscript has been deemed suitable for publication in PLOS ONE. Congratulations! Your manuscript is now with our production department. 

Kind regards, 

on behalf of

Dr. Robert Jeenchen Chen 

Academic Editor

PLOS ONE